# Water injustice in Colombia: Perceptions and realities of water quality examined through advanced machine learning

Carolina Henao-Rodríguez, Jenny-Paola Lis-Gutiérrez *

Fundación Universitaria Konrad Lorenz, Bogotá, Colombia

* jenny.lis@konradlorenz.edu.co

## Abstract

Access to safe drinking water is essential for public health. In Colombia, Resolution 2115 of 2007 mandates the use of the Water Quality Risk Index for Human Consumption (IRCA) to classify risk levels. However, a significant gap exists between objective IRCA measurements and public perception, which unfolds in the context of water injustice. This study examines the factors influencing perceptions of water quality in regions with high levels of water contamination. Based on a sample of 37,028 household heads from the 2022 Quality of Life Survey by DANE, the study applied advanced machine learning techniques, including logistic regressions with Lasso regularization and double machine learning, combining random forests and logit models. The analysis included sociodemographic factors, environmental awareness, and the environmental conditions experienced by respondents. Findings confirm the presence of water injustice in Colombia, highlighting a significant disconnect between IRCA scores and public perceptions. Additionally, perceptions of water quality are strongly influenced by visible environmental problems, such as air pollution, bad odors, and litter, suggesting that people tend to focus on more evident issues while overlooking water contamination. The results of this study highlight the need to implement policies that inform Colombians about the quality of the water they consume, in order to promote greater environmental commitment.

## 1. Introduction

According to [1], in Colombia, water pollution problems arise from multiple sources, including the discharge of untreated domestic and industrial wastewater, intensive agro-industrial practices, artisanal gold mining involving the use of mercury, and inadequate infrastructure for water treatment and monitoring.

In this regard, several studies have documented critical conditions in Colombian water bodies, such as the Apartadó River in Urabá and the Bogotá River [1,2], as well as in communities affected by monoculture and mining, including La Toma and

**Data availability statement:** Data are available at the following link: https://microdatos.dane.gov.co/index.php/catalog/793/get-microdata.

**Funding:** Funding Statement This research was funded by resources from Fundación Universitaria Konrad Lorenz.

**Competing interests:** The authors have declared that no competing interests exist.

El Tiple [3,4]. The findings of these studies revealed adverse effects on vulnerable populations, including cumulative ecological impacts and significant social and health risks, underscoring the urgent need for more effective and equitable water governance.

In this context, it is important to highlight that Colombia has a Water Quality Risk Index for Human Consumption (IRCA), which classifies water quality risk into five categories: (i)safe for human consumption (0%–5%), (ii)low risk (1%–14%), (iii) moderate risk (14.1%–35%), (iv)high risk (35.1%–80%), and (v)sanitary unviability (80.1%–100%) [5].

Meanwhile, the results Quality of Life Survey conducted by the National Administrative Department of Statistics (DANE)in 2022, revealed a mismatch between public perception and the objective IRCA measurements.

In several departments, most of the population reported never or rarely experiencing water pollution in rivers, canals, lakes, and reservoirs, while IRCA values indicate significant levels of risk.

In addition, the survey results revealed that access to drinking water in Colombia reflects the existence of water injustice. Higher-income socioeconomic groups have access to treated and safe water, whereas lower-income groups rely more heavily on precarious sources such as unpumped wells, cisterns, ponds, and rivers. Consequently, vulnerable communities are exposed to greater health risks due to the consumption of untreated water.

From this perspective, the way people perceive water pollution plays a crucial role in perpetuating environmental injustice, as it directly shapes how communities confront challenges related to this resource. In this regard, previous research has indicated that factors such as gender, age, educational attainment, and income significantly influence how individuals assess the risks associated with water quality, with direct implications for water injustice [6,7].

Against this backdrop, this study evaluates the factors that shape residents' perceptions of water quality in various Colombian regions characterized by high levels of water pollution and a context of water injustice. The sample includes 37,028 respondents represented by household heads.

To achieve the study's objective, a methodology based on logistic regression with LASSO regularization and a double machine learning approach was implemented. This methodology identifies the determinants of water contamination perception, incorporating sociodemographic variables, environmental awareness levels, and the environmental conditions to which respondents are exposed. Additionally, the study assesses the existence of a causal relationship between the reality of water contamination and public perception, offering a detailed and rigorous analysis.

This research addresses the gap highlighted by [8], who underscored the persistent challenges faced by Latin American studies in integrating multidisciplinary approaches to topics related to Political Ecology. The study adopts an environmental justice framework, a fundamental component of Political Ecology, which advocates for the equitable distribution of environmental benefits and burdens while encouraging the active participation of marginalized communities in decision-making processes.

This approach not only enables a comprehensive analysis of water perceptions in Colombia but also reveals that such perceptions are not exclusively linked to the objective conditions of water quality. Instead, they are shaped by sociodemographic, environmental, and structural factors related to access to this resource.

## 2. Literature review

### 2.1. Environmental justice

The concept of water justice is developed within the broader framework of environmental justice, understood as the equitable treatment in access, distribution, and decision-making regarding environmental benefits and burdens, regardless of race, ethnicity, or social class [9,10].

Environmental injustice, in turn, acknowledges that historically marginalized communities have disproportionately borne the burden of pollution and environmental degradation [11].

In this context [12], proposed a multidimensional approach to the study of environmental injustice, incorporating distributive justice, recognition, procedural justice, and capabilities, emphasizing the need to move beyond mere material redistribution to include effective participation and respect for cultural identities. This aligns with the work of [13] and [14], who link justice to recognition and representation within deliberative contexts.

In applied terms [15], connected environmental justice with scientific evidence of environmental harm and health inequality, highlighting the role of cities and wealthy countries in generating global impacts that disproportionately affect poorer populations. Similarly [16], argue that the classical pillars of environmental justice, distribution, participation, and recognition, should be complemented by a restorative perspective, adapted to both local and global scales.

### 2.2. Water justice

In the specific domain of water, studies such as [17] and [18] frame water justice as a distinct form of environmental injustice, where social, economic, and political factors perpetuate inequalities in access, management, and perceptions of water quality. These contributions have been instrumental in consolidating water justice as a theoretical and political tool to understand and address the unequal distribution of water resources.

Building on this theoretical foundation, recent studies have emphasized the multiscalar nature of water injustice, examining connections between socio-technical systems and historical infrastructure disparities [19]. Illustrate how historical injustices in water distribution continue to impact vulnerable communities, particularly at the intersection of water, energy, and food systems.

From a sociopolitical perspective, researchers have explored how exclusionary dynamics affect citizens' perceptions of water and their capacity to participate in decision-making processes [17] introduce the concept of "hydraulic citizenship," explaining how policies addressing environmental crises often exacerbate existing inequalities, further marginalizing vulnerable populations. This pattern aligns with structural barriers identified by [20] who highlight the inefficacy of regulations in ensuring equitable water distribution.

These exclusionary dynamics become even more pronounced in the context of large-scale hydraulic projects [18] and [21], analyze how these initiatives create tensions between communities and states, marginalizing less powerful groups in water governance. These dynamics affect not only resource distribution but also public perceptions of water quality, particularly in areas with high environmental contamination.

Moreover, structural exclusion also impacts both rural and urban poor communities [22]. Underscore the similar conditions of water marginalization faced by these groups, underscoring the need for equitable policies that prioritize the most disadvantaged populations.

In response to these challenges, intersectionality has emerged as a key theoretical tool to address inequalities in water management [23]. Advocate for active community participation to promote more inclusive and sustainable solutions, while [24] identify barriers to integrating water justice into policies addressing extreme events such as urban flooding.

## 2.3. Manifestations of water injustice: international evidence

Water injustice in Latin America appears as a deep structural issue affecting both urban and rural areas. It is shaped by geographic location, socioeconomic status, and public policy. The 2017–2018 water rationing crisis in Brasília illustrates how centralized state-private governance models reinforce inequality, disproportionately impacting low-income, less-educated, and peripheral communities through institutional violence and water alienation [7].

This pattern is evident across the region. In Peru's Ica Valley, agro-export companies, only 0.1% of users, consume 36% of available water, leaving small farmers with dry wells and no means of livelihood. In Mexico, the 2013 energy reform enabled the mining-energy sector to accumulate water rights, leading to legal and economic dispossession in Indigenous communal lands. Extractivism, as a dominant model, has intensified socio-environmental conflicts and deepened water inequities [25].

Similarly, extractive industries like mining in Cerro de San Pedro have turned ecological reserves into toxic waste sites, while hydroelectric megaprojects such as the Chixoy dam in Guatemala displaced Indigenous communities through structural violence. In urban contexts, rural to urban water transfers, such as those in Lima, strip highland communities of customary rights. Water service privatization in countries like Peru, India, and Tanzania has raised costs and reduced service quality, prompting remunicipalization [21].

Environmental risks, including urban flooding, also disproportionately affect vulnerable groups. In Santa Fe, Argentina, spatial analysis revealed systematic overexposure of marginalized communities to flood hazards, highlighting how planning policies neglect environmental equity [26].

These patterns are not isolated cases but part of a broader global trend in which policy and environmental strategies often reinforce water injustice instead of resolving it. For example [27], documents how mining activities in Finland have worsened water pollution, prompting local communities to advocate for equitable policy reforms. At the policy level [28], reveal how carbon neutrality strategies in Portugal, while environmentally driven, have unintentionally increased water access vulnerabilities for marginalized populations. Similarly [29], highlight spatial inequalities in rural Kenya, emphasizing the critical need for infrastructure design that is both socially just and context-sensitive. These cases illustrate how even progressive environmental initiatives may reproduce structural exclusion unless justice is placed at the core of water governance.

## 2.4. Factors influencing perceptions of water quality

Perceptions of water quality are shaped by a wide range of interrelated factors that go beyond objective measurements. While physicochemical indicators provide essential information, they do not fully capture how people interpret, experience, and emotionally respond to the state of water. Research has increasingly highlighted how sensory, social, cultural, psychological, and demographic elements collectively influence how water quality is perceived across diverse contexts [30,31].

In North America, a systematic review by [31] identified a set of key variables influencing perceptions among private water users. These include organoleptic attributes (such as taste, odor, and appearance), awareness of chemical and microbial contaminants, prior experiences, well infrastructure, external information sources, and demographic variables like age, gender, and income. Similarly [32], found that perceptions of drinking water safety are strongly influenced by educational level, health status, income, and social networks, often more than by objective measures.

In Latin America, these dynamics are equally evident [33]. Emphasize how socioeconomic factors, particularly education and income, directly influence perceptions of water quality and daily water-related practices, reinforcing existing inequalities [8]. Further link the lack of infrastructure in urban peripheries with heightened perceptions of water-related risk [34], meanwhile, argue that while decentralized water management solutions may offer more localized control, their effectiveness depends on context-sensitive designs to avoid reinforcing exclusion.

Previous studies have shown that environmental conditions affect the perception of water contamination. In Antofagasta, Chile, where desalinated water was introduced in 2003, [33] found that dissatisfaction with taste and concerns about quality led many households to adopt precautionary behaviors such as boiling water, even when the water met safety standards.

It is important to note that public perceptions have direct implications for the success of water policies and the adoption of sustainable practices [35] and [29]. Emphasized the importance of incorporating community perspectives into policy design to foster legitimacy, trust, and ownership. Along these lines [36], advocated for interdisciplinary educational strategies to raise awareness and promote equity in water management.

## 3. Methodology

The perception of water quality is a complex phenomenon that combines structural factors, such as infrastructure and public policies, with deeply subjective elements associated with individual experiences. In this study, we chose to employ machine learning techniques, particularly LASSO and Double Machine Learning (DML). These tools are especially useful when working with a broad and diverse set of variables, as is the case here, where socioeconomic, environmental, and demographic aspects intersect.

It is important to emphasize that these methods not only facilitate a more precise and robust selection of variables but also allow for approaching causal inferences while avoiding excessive simplifications. Previous studies, such as those by [37–39], have shown that in socio-environmental systems, patterns tend to be non-linear and difficult to anticipate; precisely for this reason, machine learning provides analytical possibilities that traditional approaches cannot achieve.

### 3.1. Data

The data used in this study were sourced from the Quality of Life Survey conducted by the National Administrative Department of Statistics (DANE) in 2022, and the Water Quality Risk Index for Human Consumption (IRCA).

The sampling design of the 2022 Quality of Life Survey employed a probabilistic, multi-stage, stratified, and clustered approach. Municipalities (primary sampling units) were selected proportionally to population size based on the 2018 census. Clusters (secondary sampling units) consisted of ten systematically selected contiguous households. The final sample included 77,400 households, distributed across urban and rural areas, with differentiated error margins and a 95% confidence interval. This design ensured national representativeness and result accuracy [40].

This study is based exclusively on secondary data from the 2022 National Quality of Life Survey (ECV), which are publicly available and fully anonymized through the microdata portal of Colombia's National Administrative Department of Statistics (DANE). The use of publicly available and anonymized secondary data does not require formal ethical approval or additional informed consent.

Table 1 summarizes the distribution of respondents, who were household heads (37,028) individuals), In the departments where IRCA levels indicated moderate or high risk. The departments include Antioquia, Boyacá, Caldas, Chocó, Huila, Magdalena, Nariño, Sucre, Tolima, Valle del Cauca, Putumayo, Amazonas, Guainía, and Vaupés (Table 1).

### 3.2. Variables

The dependent variable for this study was the perception of water contamination, assessed through the question:

*"In the past 12 months, how often have the following problems occurred in the area where your home is located: contamination in rivers, canals, lakes, and reservoirs?"*

Responses were categorized on an ordinal scale:

**Table 1. Number of Respondents by Department with High or Unsanitary Water Quality Risk Index (2022).**

| Department | Total Respondents |
| --- | --- |
| Antioquia | 3,873 |
| Boyacá | 3,209 |
| Caldas | 2,516 |
| Chocó | 2,574 |
| Huila | 2,767 |
| Magdalena | 2,296 |
| Nariño | 2,816 |
| Sucre | 2,771 |
| Tolima | 3,055 |
| Valle del Cauca | 3,152 |
| Putumayo | 2,754 |
| Amazonas | 1,788 |
| Guainía | 1,896 |
| Vaupés | 1,561 |

Note. Own elaboration. Data provided by the National Institute of Health.

1. Frequently or always

2. Never or occasionally

This indicator captures respondents' perceptions of the frequency of issues related to the contamination of water bodies in their residential areas.

Table 2 lists the predictors used in the analysis of water quality perception, grouped into sociodemographic, environmental, health, pro-environmental practices, and potable water access categories. Each category includes specific survey questions, their respective response scales, and notation codes, facilitating variable identification within the dataset.

1. **Economic and Sociodemographic Factors:** This category includes predictors such as socioeconomic status, age, gender, and ethnic self-identification to analyze differences between population groups.

Sociodemographic factors play a critical role in shaping how individuals perceive water quality, as they influence not only access to information but also lived experiences of environmental inequities. Education level, for instance, is closely linked to environmental awareness. Individuals with higher educational attainment tend to possess greater knowledge of water-related risks and standards, which in turn affects how they evaluate water safety [32].

Socioeconomic status, particularly income and housing conditions, also profoundly influences perceptions. As [33] show, low-income households frequently report poorer water quality, not necessarily because the water is chemically unsafe, but because infrastructure deficits degrade service reliability and aesthetic quality. These conditions are often compounded by geographic marginalization, such as living in urban peripheries or rural areas where public investment is limited.

2. **Environmental Perceptions:** This dimension captures perceived issues such as foul odors, garbage accumulation, and air pollution to assess the broader environmental impacts on quality of life. These perceptions are essential not only for understanding environmental discomfort but also for revealing how external environmental cues shape attitudes toward other public services, including water.

**Table 2. Predictors of Water Quality Perception in Colombia (2022).**

| Aspect Measured | Question/Category | Notation Code |
|---|---|---|
| **Economic and Sociodemographic Factors** | Socioeconomic stratum of electricity service: 1. Very Low; 2. Low; 3. Lower Middle; 4. Middle; 5. Upper Middle; 6. High; 8. Generator; 9. Unknown or lacks payment receipt; 0. Informal/unofficial service | P8520S1A1 |
| | Age: How old are you? | P6040 |
| | Gender at birth: 1. Male; 2. Female | P6020 |
| | Based on your culture, community, or physical traits, do you identify as: 1. Indigenous; 2. Romani (Gitano); 3. Raizal from the Archipelago of San Andrés, Providencia, and Santa Catalina; 4. Palenquero from San Basilio; 5. Black, Mulatto, Afro-Colombian; 6. None of the above | P6080 |
| | In the past 12 months, did you or anyone in your household worry about not having enough food due to lack of resources? 1. Yes; 2. No; 3. Unsure/No response | P3516S1 |
| | Marital Status: 1. Not married, cohabiting < 2 years; 2. Not married, cohabiting ≥ 2 years; 3. Widowed; 4. Separated/Divorced; 5. Single; 6. Married | P5502 |
| **Location** | Department:5. Antioquia; 8. Atlántico; 11. Bogotá; 13. Bolívar; 15. Boyacá; 17. Caldas; 18. Caquetá; 19. Cauca; 20. Cesar; 23. Córdoba; 25. Cundinamarca; 27. Chocó; 41. Huila; 44. La Guajira; 47. Magdalena; 50. Meta; 52. Nariño; 54. Norte de Santander; 63. Quindío; 66. Risaralda; 68. Santander; 70. Sucre; 73. Tolima; 76. Valle del Cauca; 81. Arauca; 85. Casanare; 86. Putumayo; 88. San Andrés; 91. Amazonas; 94. Guainía; 95. Guaviare; 97. Vaupés; 99. Vichada | P1_DEPAR-TAMENTO |
| **Environmental Perceptions** | In the past 12 months, how often did the following issues occur in your area: Foul odors from outside? 1. Never; 2. Occasionally; 3. Often; 4. Always | P5661S2 |
| | In the past 12 months, how often did the following issues occur in your area: Garbage in the streets? 1. Never; 2. Occasionally; 3. Often; 4. Always | P5661S3 |
| | In the past 12 months, how often did the following issues occur in your area: Air pollution? 1. Never; 2. Occasionally; 3. Often; 4. Always | P5661S4 |
| | Do you consider the available water sufficient? 1. Yes; 2. No | P3166 |
| **Subjective Health Perception** | Overall, how satisfied are you with your current health? 10. Completely satisfied; 0. Completely dissatisfied | P1897 |
| **Pro-environmental Practices** | What practices does your household use to conserve water: Reusing water? 1. Yes; 2. No | P5012S4 |
| | What practices does your household use to conserve water: Collecting rainwater? 1. Yes; 2. No | P5012S5 |
| | What practices does your household use to conserve water: Using low-consumption water tanks? 1. Yes; 2. No | P5012S6 |
| | What practices does your household use to conserve energy: Unplugging electrical devices? 1. Yes; 2. No | P5012S8 |
| **Potable Water Access and Infrastructure** | Does water reach your home 24 hours a day, seven days a week? 1. Yes; 2. No | P5047 |
| | Does your household pay for water services? 1. Yes; 2. Yes, included in rent; 3. No, but service is provided; 4. No service available | P792 |
| | Main source of water for food preparation: 1. Public aqueduct; 2. Community or rural aqueduct; 3. Pumped well; 4. Non-pumped well or cistern; 5. Rainwater; 6. River, stream, spring; 7. Public tap; 8. Tanker truck; 9. Water vendor; 10. Bottled or bagged water | P8530 |

Note. Own elaboration. Data sourced from [40].

Perceptions of water quality are strongly influenced by broader environmental concerns such as air pollution, waste, flooding, and climate change, forming a cognitive and emotional ecosystem in which accumulated environmental experiences affect trust in water resources. Studies in both rural and urban contexts [26,41] show that visual and symbolic factors, such as visible waste or extreme weather events, intensify perceptions of vulnerability and distrust, even when technical indicators of water quality are adequate.

Furthermore, exposure to negative environmental messages in media or social networks can amplify these perceptions [38].

3. **Subjective Health Perception:** Assesses satisfaction with general health on a scale from 0 to 10, reflecting individual well-being. At this point, it is important to highlight that when individuals perceive themselves as healthy, they may tend to assume that the water they consume is safe, without necessarily considering the actual levels of contaminants or linking their health issues to potential water contamination. As demonstrated by the studies of [42,43].

4. **Pro-environmental Practices:** This category includes variables such as water reuse, rainwater collection, use of low-flush toilets, and unplugging electronic devices to measure sustainable household habits. Although a broader range of practices was initially considered, such as waste separation, use of energy-saving light bulbs, turning off lights, efficient ironing or avoiding ironing, and installation of water-saving devices in showers and faucets, some of these did not show statistical significance in preliminary models and were therefore excluded from the analysis. Only the variables with empirical relevance were retained, as they more clearly captured sustainable behavior at the household level.

Environmental awareness plays a crucial role in shaping perceptions of water quality, as it enhances individuals' ability to identify ecological risks and relate them to their immediate surroundings. In this context, the study by [44], based on Eurobarometer data and the WISE system of the European Environment Agency, demonstrated that perceived water-related risks are significantly influenced by environmental knowledge, participation in pro-environmental activities, and conscious consumption practices. These factors increase sensitivity to actual environmental changes, aligning individual perceptions with the objective condition of the water environment.

5. **Potable Water Access and Infrastructure:** Examines water supply, primary sources for drinking and cooking, and sufficiency of services. Water access infrastructure is a key factor in shaping perceptions of water quality. As demonstrated by [31] in their systematic review of private water users in Canada and the United States, the structural conditions of water collection systems, such as their age, maintenance, and the type of technology used, directly influence how users perceive the safety of the water they consume.

### 3.3. Algorithms

#### 3.3.1. Penalized Logistic Regression (LASSO).
The LASSO method, introduced by [45], combines logistic regression with penalization for variable selection and regularization. The loss function is defined as:

$$\text{Min}\left(\frac{1}{n}\sum_{i=1}^{n}\left(y_i\, log(\widetilde{p_i}) + (1-y_i)log(1-\widetilde{p_i})\right) + \lambda\sum_{j=1}^{p}|\beta_j|\right)$$

(1)

Where:

- $\widetilde{p_i}$: Predicted probability for observation iii,

- $y_i$: Binary dependent variable (0 or 1),

- $\beta_j$: Estimated coefficients,

- $\lambda$: Regularization parameter controlling penalization.

The optimal λ\lambdaλ value was selected using kkk-fold cross-validation in Stata.

**3.3.2. Double Machine Learning (DML).** Double Machine Learning estimates causal effects by separating covariate modeling from treatment effects [46]. The approach comprises two stages:

1. **Stage One:** Models dependent variables and controls using random forests and logistic regression to adjust non-parametric effects.

2. **Stage Two:** Uses residuals from the first stage to estimate causal parameters with adjusted linear or logistic regression.

   The DML framework for causal estimation is defined as:

$$\theta = \frac{1}{n} \sum\nolimits_{i=1}^{n} [(y_i - m(X_i))(D_i - k(X_i))]$$

(2)

Where:

- $y_i$: Observed outcome (dependent variable), representing the perception of water quality relative to IRCA values.

- $D_i$: Binary treatment variable ($ccc$), set to 1 if contam_valor≥65, and 0 otherwise.

- $X_i$: Covariates, including demographic, socioeconomic, environmental, and pro-environmental factors (see Table 1).

- $m(X_i)$ and $k(X_i)$: Adjusted predictions of non-parametric functions

# 4. Results

This section provides a detailed analysis of findings related to water injustice in Colombia and the determinants influencing the perception of water quality. The results reveal how socioeconomic and environmental inequalities shape public perceptions of water contamination, often misaligned with objective indicators like the Water Quality Risk Index for Human Consumption (IRCA). Additionally, the analysis highlights disparities in access to potable water and the influence of factors such as pro-environmental practices, health conditions, and sociodemographic characteristics on perceptions of water-related risks.

## 4.1. Water injustice in Colombia

Table 3 illustrates how respondents perceive the frequency of contamination in rivers, canals, lakes, and reservoirs across various Colombian departments over the past 12 months. These perceptions are presented alongside the Water Quality Risk Index for Human Consumption (IRCA) and its corresponding classifications, providing a way to compare the community's subjective experience with objective water quality measurements.

The data in Table 3 illustrate a significant disconnect between objective IRCA measurements and public perception of contamination in water bodies. Departments with IRCA ratings classified as high risk, such as Nariño (IRCA=46), Guanía (IRCA=37), and Vaupés (IRCA=51), show that over 72 percent of respondents perceive contamination as occurring "never" or "sometimes," significantly underestimating the objective risk.

In Colombia, access to potable water is deeply inequitable, with disparities tied to socioeconomic strata. Higher- and middle-income strata (3, 4, 5, 6) rely predominantly on public water systems, with between 88 and 93 percent of households connected to treated and regulated sources ensuring access to treated and safe water. In contrast, lower-income groups (strata 0, 1, and 2) have much lower access to public water networks, ranging from 14.9 to 61,7 percent, and are more dependent on precarious sources such as wells, cisterns, and rivers, which account for up to 57 percent of supply in stratum 0, and 33 percent in stratum 1, thereby increasing exposure to health risks. Communal water systems are also

Table 3. Perception of water quality and the water quality risk index for human consumption (IRCA) by Colombian departments (2022).

| Department | "Over the past 12 months, how often have the following issues occurred in your area: contamination in rivers, canals, lakes, and reservoirs?" (Responses regarding the perceived frequency of contamination in water bodies are expressed as percentages). | | | | IRCA | IRCA Classification |
|---|---|---|---|---|---|---|
| | Never | Sometimes | Many Times | Always | | |
| Antioquia | 71.5 | 22.6 | 3.4 | 2.5 | 16 | Moderate risk |
| Boyacá | 89.7 | 7.9 | 1.2 | 1.1 | 24 | Moderate risk |
| Caldas | 90.0 | 7.0 | 1.7 | 1.3 | 35 | Moderate risk |
| Chocó | 43.1 | 34.9 | 9.9 | 12.1 | 25 | Moderate risk |
| Huila | 87.3 | 10.3 | 1.2 | 1.2 | 21 | Moderate risk |
| Magdalena | 66.5 | 22.1 | 4.0 | 7.4 | 15 | Moderate risk |
| Nariño | 78.8 | 11.3 | 3.6 | 6.4 | 46 | High risk |
| Sucre | 83.6 | 11.4 | 3.0 | 2.1 | 18 | Moderate risk |
| Tolima | 86.5 | 9.1 | 2.8 | 1.5 | 18 | Moderate risk |
| Valle del Cauca | 75.5 | 15.7 | 4.7 | 4.1 | 22 | Moderate risk |
| Putumayo | 57.9 | 35.4 | 3.5 | 3.2 | 21 | Moderate risk |
| Amazonas | 65.5 | 27.3 | 6.2 | 1.0 | 17 | Moderate risk |
| Guainía | 71.7 | 23.5 | 2.3 | 2.5 | 37 | High risk |
| Vaupés | 77.5 | 19.5 | 2.3 | 0.6 | 51 | High risk |

**Note:** Own elaboration using data from [40] and the National Institute of Health.

more common in lower strata, reaching up to 25 percent in stratum 1, but remain marginal in higher strata, where their use falls below ten percent (see Table 4).

These disparities not only reflect unequal access to water infrastructure but also perpetuate structural marginalization that reinforces socioeconomic inequities.

The findings highlight the presence of structural water injustice in Colombia. The gap between public perception and technical water quality indicators, particularly in departments classified as high risk, reveals a critical information void that obscures the real dangers faced by the most vulnerable communities.

This disconnection is further deepened by stark inequalities in access to safe drinking water, shaped by socioeconomic stratification. While wealthier households benefit from formal and regulated water systems, low-income families rely on unsafe sources such as untreated wells and rivers, which heightens their exposure to health risks and limits their ability to demand fair environmental protections.

Taken together, the empirical evidence confirms that social, economic, and territorial conditions determine not only access to water but also how its quality is perceived, perpetuating exclusionary patterns that must be addressed through a water justice perspective.

## 4.2. Determinants of perceived water quality in Colombia (2022)

### 4.2.1. Logistic regression with LASSO regularization.
For the LASSO-regularized logistic regression model, a dependent variable called "perception of water contamination" was constructed as a binary variable (1 = "often" or "always"; 0 = "never" or "sometimes"). The model demonstrated strong predictive power in assessing perceptions of water quality in Colombia, as evidenced by the Area Under the Receiver Operating Characteristic Curve (AUC). In both the training dataset (n = 19,319) and the validation dataset, the model achieved an AUC of 0.89, indicating a high level of

**Table 4. Distribution of population access to water sources by socioeconomic stratum in departments with high or sanitarily unviable IRCA (2022).**

| Stratum | Public Water System | Communal Water System | Well with Pump | Well without Pump | Rainwater | River/ Creek/ Stream | Bottled Water |
|---|---|---|---|---|---|---|---|
| 0 | 14.9 | 21.5 | 9.9 | 5.0 | 28.4 | 13.8 | 0.6 |
| 1 | 37.0 | 25.2 | 6.5 | 4.8 | 12.8 | 9.2 | 0.1 |
| 2 | 61.6 | 21.4 | 5.4 | 2.6 | 0.4 | 6.8 | 0.3 |
| 3 | 86.5 | 3.7 | 5.9 | 0.2 | 0.1 | 0.6 | 0.5 |
| 4 | 92.6 | 3.2 | 1.2 | 0.1 | 0.1 | 2.2 | 0.1 |
| 5 | 93.0 | 4.9 | 0.4 | 0.4 | 0.0 | 0.0 | 0.0 |
| 6 | 78.8 | 6.2 | 3.4 | 4.8 | 0.0 | 2.1 | 0.0 |

**Note:** All values in the table are expressed as percentages (%) and represent the proportion of the population within each socioeconomic stratum that reported using the respective water source. Own elaboration based on data from the National Quality of Life Survey by [40].

accuracy. These results confirmed the robustness of the model in identifying the factors that shape perceptions of water quality in Colombia.

LASSO regularization made it possible to select only the most relevant variables, reduce the risk of overfitting and multicollinearity, and ensure the interpretability of the results

Among environmental predictors, variables reflecting perceived problems such as air pollution (P5661S4), foul odors (P5661S2), and visible waste (P5661S3) exhibit significant negative coefficients. This indicates that as perceptions of these issues increase, the likelihood of reporting high levels of water contamination decreases (Table 5). These findings suggest that respondents may focus on specific environmental problems, potentially reducing their awareness of others, such as water contamination. The results also highlight potential limitations in respondents' overall environmental awareness or differences in the visibility and understanding of various environmental issues.

This trend may be explained by a selective attention effect, where individuals focus on more immediate, visible, or sensory environmental cues, thereby diminishing their awareness of less tangible threats, such as water quality [26,41] describe similar dynamics, in which environmental distress is cognitively distributed across several issues, leading to perceptual prioritization of some environmental problems over others.

The IRCA variable shows a negative coefficient ($\beta = -0.2746$), indicating that higher water risk scores are associated with lower perceptions of contamination. This result may reflect a disconnect between objective conditions and subjective perceptions, where individuals either fail to associate higher IRCA values with contamination risks or lack awareness of the significance of these indices (Table 5). In the Colombian context, this disconnect may stem from sociopolitical dynamics that perpetuate a lack of information and community participation in water governance [18,20].

In terms of pro-environmental practices, the results indicate that certain sustainable behaviors correlate with lower perceptions of water contamination. For instance, the use of low-flow toilets (1.P5012S6, coef. = −0.0932154), and water reuse (1.P5012S4, coef. = −0.0056787), display significant negative coefficients. This suggests that households adopting proactive environmental measures tend to perceive fewer water quality issues, potentially due to a higher level of awareness and confidence in their ability to mitigate environmental risks (Table 5). This supports the findings of [44], who show that environmental engagement is associated with greater ecological trust.

Conversely, individuals collecting rainwater as an alternative practice (1.P5012S5, coef. = 0.0758591) are more likely to perceive water contamination problems. This may stem from the need to supplement traditional water sources due to perceived insufficiency or poor quality. Additionally, these individuals may have greater direct exposure to environmental

**Table 5. Logistic Regression Results with LASSO Regularization.**

| Variable | logit |
| --- | --- |
| _cons | −1.989267 |
| P5661S4 | |
| 1 | −0.6428945 |
| P5661S2 | |
| 1 | −0.5732492 |
| P5661S3 | |
| 1 | −0.4415709 |
| IRCA | −0.2746187 |
| P8520S1A1 | |
| 1 | 0.2165034 |
| 1.P8530 | −0.1822817 |
| 1.P5047 | 0.1312053 |
| P8520S1A1 | |
| 8 | 0.1166384 |
| P5661S3 | |
| 2 | 0.1035229 |
| P1897 | −0.100521 |
| 1.P5012S6 | −0.0932154 |
| P6080 | |
| 6 | −0.0875754 |
| P5661S2 | |
| 4 | 0.0865174 |
| P8520S1A1 | |
| 3 | −0.0808111 |
| 1.P5012S5 | 0.0758591 |
| P8520S1A1 | |
| 5 | −0.0720118 |
| 9 | 0.0677313 |
| P5661S4 | |
| 4 | 0.0656007 |
| P6040 | −0.0625291 |
| P6080 | |
| 5 | 0.0612442 |
| 1.P3166 | −0.0490439 |
| P5661S4 | |
| 3 | 0.0472912 |
| P5661S2 | |
| 3 | 0.0400895 |
| P8520S1A1 | |
| 4 | −0.0396612 |
| 1.P3516S1 | −0.0303174 |
| P6080 | |
| 2 | −0.029777 |
| P5069 | |
| 1 | −0.0278143 |

*(Continued)*

**Table 5.** (Continued)

| Variable | logit |
|---|---|
| P8520S1A1 | |
| 6 | −0.0250513 |
| P792 | |
| 3 | 0.0239593 |
| P5661S3 | |
| 4 | 0.0213286 |
| P5502 | |
| 4 | 0.0202165 |
| P5069 | |
| 4 | 0.0127189 |
| 3 | −0.0109199 |
| P6080 | |
| 4 | −0.0093264 |
| P792 | |
| 1 | −0.0087834 |
| 1.P5012S8 | 0.0075838 |
| P5069 | |
| 5 | 0.0065606 |
| P5502 | |
| 6 | −0.0060278 |
| 1.P5012S4 | −0.0056787 |
| P5502 | |
| 5 | −0.0056237 |

Note: Own elaboration using Stata [47].

conditions, allowing them to identify visible signs of contamination, such as turbidity, odors, or debris in collected rainwater (Table 5).

The analysis of water service payment schemes (P792) reveals differences in perceptions based on payment methods. Households receiving free services (category 3, coef. = 0.0239593) are more likely to perceive lower water quality, potentially due to associations with inadequate infrastructure or irregular maintenance. In contrast, households that pay for water services (category 1, coef. = −0.0087834) exhibit lower contamination perceptions, possibly reflecting greater trust in formal supply systems and their quality (Table 5). Validating the observations of [31], formal and regulated access strengthens the perception of safety.

The water sources used for food preparation (P8530) reflect the diversity of access to water across the country. Households relying on public water systems (Category 1, coef. = −0.1822817) are less likely to perceive water contamination (Table 5).

The empirical results of the study support and extend existing evidence on the role of sociodemographic factors in shaping perceptions of water quality. Specifically, higher socioeconomic strata (strata 5 and 6) are less likely to perceive contamination (P8520S1A1, coef. = − 0.0720118, and coef. = −0.0250513), while lower-income households (strata 0 and 1) report higher perceptions of water contamination. These findings are consistent with [33], who argue that poor water quality perception among low-income households is not always due to chemical contamination, but rather to infrastructure deficits that degrade the reliability and aesthetic quality of water services.

Moreover, ethnic self-identification (P6080) and age (P6040) further reflect the influence of cultural and generational identities on environmental awareness. Afro-descendant individuals report higher contamination perceptions, potentially linked to historical or geographic exposure to unsafe water sources. In contrast, other ethnic groups show lower perceptions of contamination. This reinforces the view expressed by [32,33] that territorial marginalization and access to environmental resources critically shape lived environmental experiences.

Additionally, age is negatively associated with contamination perception ($\beta = 0.0625291$), suggesting that older individuals are less likely to perceive water contamination. This may be attributed to differences in perception thresholds or reduced exposure to contemporary environmental education, reflecting generational gaps in ecological awareness, again aligning with [32] arguments.

Regarding age, results show that as respondents age, the likelihood of perceiving water contamination decreases ($\beta = -0.0427$). This pattern may be associated with differences in environmental perception thresholds, lower exposure to contemporary educational campaigns, or reduced interaction with media emphasizing water quality concerns (Table 5).

Health perception ($P1897 = -0.100521$) also emerges as a key factor. Individuals reporting higher satisfaction with their health tend to perceive less contamination (Table 5), a finding that aligns with [42,43]. These authors argue that positive subjective health reinforces assumptions of environmental safety, potentially reducing critical engagement with actual contamination risks.

**4.2.2. Double machine learning analysis.** The analysis of results obtained through the Double Machine Learning (DML) method, applied to evaluate the causal relationship between the IRCA index and the perception of water contamination, provides valuable insights into the existence and direction of this relationship.

From a technological perspective, the use of advanced tools such as Double Machine Learning (DML) and LASSO in this study reinforces findings by [37] regarding the capacity of these techniques to identify complex patterns and nonlinear relationships among demographic, socioeconomic, and environmental variables.

Logistic regression (logit) and random forest (rforest) models were used to estimate the conditional expectations of y|X and D|X, as this approach provides greater flexibility in modeling non-linear relationships. To minimize estimation bias, a cross-validation procedure with four folds (k-folds(4)) was applied.

The impact of D on Y was assessed by testing different model combinations and selecting the one with the lowest mean squared error (MSE). The estimates remained robust across various specifications, including the model with the lowest MSE and the so-called "shortstack" model, reinforcing the reliability of the findings. Furthermore, the consistent statistical significance across multiple configurations suggests that the estimated relationship is strong

The model coefficients indicate the causal relationship between the water risk level (IRCA) and the perception of contamination. The analysis identifies whether a disconnect exists between objective reality (IRCA) and subjective perceptions. In the model with the lowest MSE, the coefficient of IRCA on y was $c = -0.0812918$ with a standard error of 0.004. This result is statistically significant at the 1% significance level (Table 6).

**Table 6. Shortstack double machine learning model results.**

| y-E[yX] =y-Y_water_ss_1  D-E[DX] =D-D_c_ss_1 | | | | Number of obs= 27.872 | | |
|---|---|---|---|---|---|---|
| y | Coefficient | Robust std. err. | z | P>z | [95% conf. interval] | |
| c | −0.0812918 | 0.0049012 | −16.59 | 0 | −0.0908979 | −0.0716856 |
| _cons | 0.0000517 | 0.0021313 | 0.02 | 0.981 | −0.0041254 | 0.0042289 |
| | | | | | | |
| Water | Coefficient | Robust std. err. | z | P>z | [95% conf. interval] | |
| c | −0.0812918 | 0.0049012 | −16.59 | 0 | −0.0908979 | −0.0716856 |
| _cons | 0.0000517 | 0.0021313 | 0.02 | 0.981 | −0.0041254 | 0.0042289 |

Note: Own elaboration using Stata [47].

The negative coefficient c indicates that as the IRCA index (a water quality risk indicator) increases, the likelihood of individuals perceiving water contamination decreases. This result highlights a disconnect between subjective perceptions and objective conditions, where people may not directly associate higher IRCA values with water quality issues or may lack awareness of the risks these indices represent. This reveals a structural disconnect between objective risk conditions and citizens' subjective perceptions, which could hinder the formulation of effective public policies unless both the technical and perceptual components of the problem are simultaneously addressed. This gap has been widely discussed by scholars such as [17,21,35], who emphasize the need for participatory and contextually adapted strategies to achieve equitable water governance.

## 5. Conclusions

This study reveals a significant disconnect between public perceptions of water quality and the objective conditions measured through the Water Quality Risk Index (IRCA) in Colombia. This gap highlights how socioeconomic, cultural, and environmental factors shape subjective perceptions, even when they diverge from technical indicators.

The research addresses a critical gap in the literature by combining advanced analytical approaches, such as Double Machine Learning (DML) and penalized logistic regression (LASSO), to compare perceptions with the reality of water quality. While previous studies have examined water injustice mainly from a sociopolitical perspective [17,18], this work provides a robust quantitative analysis that captures the complex interactions among demographic, socioeconomic, and environmental variables. Moreover, the use of machine learning tools complements traditional approaches by identifying non-linear patterns and causal relationships that have been largely unexplored in the Colombian context.

The findings emphasize that the most vulnerable communities, particularly those in lower socioeconomic strata, face not only inequalities in access to safe drinking water but also structural marginalization that limits their ability to adequately interpret the risks associated with water quality. Furthermore, communities simultaneously exposed to issues such as air pollution, visible waste accumulation, and foul odors tend to selectively prioritize these environmental problems, which may distort their perceptions of water quality.

Indeed, visible environmental problems, such as waste accumulation, unpleasant odors, and air pollution, are negatively associated with perceptions of water contamination, suggesting a pattern of selective environmental attention. These findings reinforce previous evidence on environmental cognition [26,30,41].

The results have direct implications for public policy and water governance. First, the disconnect between perception and reality underscores the need to improve public communication strategies. Second, the results highlight the urgency of addressing structural inequalities in access to potable water. This requires prioritizing investments in water infrastructure for marginalized communities and ensuring their active participation in decision-making processes. Third, the negative perceptions identified, particularly among lower-income groups, may pose a barrier to the implementation of water policies unless citizens' concerns are integrated into the design of these initiatives. Incorporating public perceptions into policy formulation can strengthen community ownership and enhance the social acceptance of water management programs.

While this study offers a robust quantitative analysis of water quality perceptions in Colombia, it is important to acknowledge its limitations regarding the inclusion of communities in direct contact with natural water bodies. In particular, the dataset used (DANE's 2022 Quality of Life Survey) does not provide disaggregated data for riverside populations, such as those living in the Atrato River basin.

Given that Colombia has been a pioneer in the legal recognition of the Rights of Nature, as demonstrated by the constitutional personhood granted to the Atrato River, future research should consider incorporating qualitative and participatory methodologies in regions such as Chocó. These approaches would allow for a deeper understanding of how legal frameworks related to environmental rights intersect with the lived experiences and local perceptions of marginalized communities in water-conflicted territories. Ethnographic studies, community-based participatory research, and interviews with Afro-Colombian and Indigenous groups in these areas could provide essential insights to complement national-level statistical models and enrich the study of water justice.

This study opens multiple avenues for future research. First, longitudinal studies are needed to evaluate how perceptions evolve over time and in response to changes in water policies and environmental conditions. Second, expanding the analysis to rural and urban contexts in other Latin American countries would provide valuable comparisons of perception patterns and their relationship with water injustice dynamics.

## Author contributions

**Conceptualization:** Carolina Henao-Rodríguez.

**Data curation:** Carolina Henao-Rodríguez.

**Formal analysis:** Carolina Henao-Rodríguez, Jenny-Paola Lis-Gutiérrez.

**Investigation:** Carolina Henao-Rodríguez.

**Methodology:** Carolina Henao-Rodríguez, Jenny-Paola Lis-Gutiérrez.

**Writing – original draft:** Carolina Henao-Rodríguez.

**Writing – review & editing:** Carolina Henao-Rodríguez, Jenny-Paola Lis-Gutiérrez.

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
