## [Decision Letter · Decision Letter 0]

26 Jun 2025

PONE-D-25-04781Water Injustice in Colombia: Perceptions and Realities of Water Quality Examined Through Advanced Machine LearningPLOS ONE

Dear Dr. Lis-Gutiérrez,

Thank you for submitting your manuscript to PLOS ONE. After careful consideration, we feel that it has merit but does not fully meet PLOS ONE’s publication criteria as it currently stands. Therefore, we invite you to submit a revised version of the manuscript that addresses the points raised during the review process. Please submit your revised manuscript by Aug 10 2025 11:59PM. If you will need more time than this to complete your revisions, please reply to this message or contact the journal office at plosone@plos.org . Please include the following items when submitting your revised manuscript:

We look forward to receiving your revised manuscript.

Kind regards,

Diogo Guedes Vidal, PhD

Academic Editor

PLOS ONE

Journal Requirements:

2. Please note that PLOS ONE has specific guidelines on code sharing for submissions in which author-generated code underpins the findings in the manuscript. In these cases, we expect all author-generated code to be made available without restrictions upon publication of the work. 

Please review our guidelines at https://journals.plos.org/plosone/s/materials-and-software-sharing#loc-sharing-code and ensure that your code is shared in a way that follows best practice and facilitates reproducibility and reuse.

“Funding Statement

This research was funded by resources from Fundación Universitaria Konrad Lorenz”

6. We note that Figure 1 in your submission contain map images which may be copyrighted. All PLOS content is published under the Creative Commons Attribution License (CC BY 4.0), which means that the manuscript, images, and Supporting Information files will be freely available online, and any third party is permitted to access, download, copy, distribute, and use these materials in any way, even commercially, with proper attribution. For these reasons, we cannot publish previously copyrighted maps or satellite images created using proprietary data, such as Google software (Google Maps, Street View, and Earth). For more information, see our copyright guidelines: http://journals.plos.org/plosone/s/licenses-and-copyright.

1) You may seek permission from the original copyright holder of Figure 1 to publish the content specifically under the CC BY 4.0 license.  

2) If you are unable to obtain permission from the original copyright holder to publish these figures under the CC BY 4.0 license or if the copyright holder’s requirements are incompatible with the CC BY 4.0 license, please either i) remove the figure or ii) supply a replacement figure that complies with the CC BY 4.0 license. Please check copyright information on all replacement figures and update the figure caption with source information. If applicable, please specify in the figure caption text when a figure is similar but not identical to the original image and is therefore for illustrative purposes only.

Reviewers' comments:

Reviewer's Responses to Questions

**Comments to the Author**

1. Is the manuscript technically sound, and do the data support the conclusions?

Reviewer #1: Yes

Reviewer #2: No

2. Has the statistical analysis been performed appropriately and rigorously? 

Reviewer #1: I Don't Know

Reviewer #2: I Don't Know

3. Have the authors made all data underlying the findings in their manuscript fully available?

Reviewer #1: Yes

Reviewer #2: No

4. Is the manuscript presented in an intelligible fashion and written in standard English?

Reviewer #1: Yes

Reviewer #2: No

5. Review Comments to the Author

Reviewer #1: Given the importance of implementing policies that bring perceptions closer to reality and of combating water injustice in Colombia, this work deserves to be published. However, as Colombia is a country whose ancestral influence has facilitated the implementation of the Rights of Nature and the Atrato River has even been granted legal rights, this work would be more complete with data from the populations that are in direct contact with the river, namely perceptions and water-related injustices.

The astract should be revised, shortening the text and at the same time clarifying the work, where it started from, the methodology and the results.

In point 2, it would be interesting to list the authors who helped define water justice.

The example you provide about mining in Finland should be supplemented with examples from researchers working on the ground in Latin America.

The examples presented in the aspect measured on Pro-environmental Practices are very poor. For example, it doesn't talk about waste treatment, electrical and electronic equipment, the preservation of natural resources, etc.

Reviewer #2: Dear authors, both similarity and AI seem to have been very extensively used in the manuscript, easily recognizable.

General comments, then:

The abstract is not in line with an assertive and clear manuscript relating context methods, findings and implications.

The introduction section lacks references contextualization and in fact is very general.

All the mentioned values would require literature support and further contextualization.

Paragraphs are disconnected and separated from one another.

The literature review section requires streamlining and further interlinkage.

The methods section also requires assertiveness and literature support no listing.

The absence of references before equations would mean complete originality, clearly not the case.

The results section require significant improvements, upper letter no to be used, why add % next to each value_ Etc.

It is not clear what is being presented.

Why add columns where the value never changes?

The discussion section would have to be merged with the results section supporting or opposing the results presented.

The conclusions section would have to start by clarifying the context, the methods used, the main findings, the implications, the limitations and future prospects.

This is clearly not the way to indicate data used as authors would have to further specify.

Availability of data and materials. All data used in the paper are publicly available at https://microdatos.dane.gov.co/index.php/catalog/793/get-microdata

6. PLOS authors have the option to publish the peer review history of their article (what does this mean? ). If published, this will include your full peer review and any attached files.

**Do you want your identity to be public for this peer review?** For information about this choice, including consent withdrawal, please see our Privacy Policy .

Reviewer #1: No

Reviewer #2: No

---

## [Author Response · Author response to Decision Letter 1]

4 Aug 2025

Dear Editor and reviewers

We would like to express our sincere gratitude to the reviewers for their careful reading of our manuscript and for providing insightful comments and suggestions. We have carefully addressed all the points raised and revised the manuscript accordingly. Below, we provide a detailed, point-by-point response. Reviewer comments are presented in italicized quotes, followed by our responses in regular text.

Reviewer #1

• “ Given the importance of implementing policies that bring perceptions closer to reality and of combating water injustice in Colombia, this work deserves to be published. However, as Colombia is a country whose ancestral influence has facilitated the implementation of the Rights of Nature and the Atrato River has even been granted legal rights, this work would be more complete with data from the populations that are in direct contact with the river, namely perceptions and water-related injustices.”

We appreciate this valuable observation. We fully agree that the experiences of communities in direct contact with water bodies such as the Atrato River are fundamental for analyzing water justice. Although the present study was based on microdata from the DANE Quality of Life Survey, which does not include specific disaggregation for riverside communities along the Atrato, we acknowledge this limitation and propose to include it in the conclusions section. As a methodological suggestion for future work, we recommend the integration of ethnographic and participatory data in regions such as Chocó, where the legal recognition of the Atrato River allows for the observation of the intersection between environmental rights and local perceptions.

• “The astract should be revised, shortening the text and at the same time clarifying the work, where it started from, the methodology and the results.”

In response, we have revised the abstract to improve its clarity and conciseness. The updated version clearly outlines the motivation behind the study, the methodological approach, and the main findings.

• “In point 2, it would be interesting to list the authors who helped define water justice.”

In response, we have revised Section 2 to explicitly include key authors who have contributed to the conceptual development of water justice. We incorporated foundational works that link water justice to broader environmental justice frameworks, including Bullard (1990), Schlosberg (2007), Sen (2009), and Fraser (2009), as well as more recent contributions by David and Hughes (2024), Rodríguez-Labajos et al. (2015), and Stephens (2007). These references highlight the multidimensional nature of water justice, encompassing distributive, procedural, and recognition-based dimensions, along with the structural and political conditions that shape unequal access to water.

• “The example you provide about mining in Finland should be supplemented with examples from researchers working on the ground in Latin America.”

In response, we have complemented the example of mining in Finland with several grounded case studies from Latin America, offering a more regionally contextualized analysis of water injustice. For instance, we highlight the case of Cerro de San Pedro (Mexico), where Canadian mining operations turned an ecological reserve into a toxic waste site, displacing local communities (Boelens et al., 2018). We also reference Jacobo-Marín (2025), who documents how Mexico's 2013 energy reform enabled the legal appropriation of water rights by the mining-energy sector, deepening dispossession in Indigenous communal lands. Furthermore, we include a case from Santa Fe, Argentina, where Bosisio & Moreno-Jiménez (2022) show how urban flooding systematically impacts marginalized communities, revealing spatial patterns of hydrosocial inequality.

• “The examples presented in the aspect measured on Pro-environmental Practices are very poor. For example, it doesn't talk about waste treatment, electrical and electronic equipment, the preservation of natural resources, etc.”

In response, we have revised the corresponding section to clarify that an initially broader set of sustainable household practices was included. These practices encompassed: waste separation, use of energy-saving light bulbs, switching off lights, ironing efficiently or not at all, unplugging electrical devices, and installing water-saving devices in showers and faucets. However, after applying preliminary machine learning models, several of these variables were not statistically significant and were therefore excluded from the final model. The analysis retained only those variables that showed robust associations with the outcomes, specifically: water reuse, rainwater harvesting, use of low-consumption toilet tanks, and unplugging of electrical appliances. These variables more accurately capture sustainable household behaviors.

Reviewer #2:

• “Dear authors, both similarity and AI seem to have been very extensively used in the manuscript, easily recognizable.”

As part of the revision process, we conducted a full similarity check using Turnitin. The results confirmed that the overall similarity index is below 10%, remaining well within acceptable academic standards. The report also shows no indication of AI generated content. For transparency, we have included a screenshot of the Turnitin report.

We would also like to clarify that ChatGPT was used solely for language style refinement and minor editorial corrections, without contributing to the substantive content or analysis of the manuscript. All substantive content, analysis, and academic writing were carried out independently by the authors.

• “The abstract is not in line with an assertive and clear manuscript relating context methods, findings and implications.”

In response, we have thoroughly revised the abstract to better align it with the structure and content of the manuscript. The updated version now clearly establishes the contextual background of water injustice in Colombia, outlines the methodology, specifically the use of advanced machine learning techniques, and presents the main findings. These include the gap between IRCA scores and public perception, and the influence of visible environmental factors on water quality perceptions. Finally, the abstract now explicitly highlights the implications for public policy and water governance.

• “The introduction section lacks references contextualization and in fact is very general. All the mentioned values would require literature support and further contextualization. Paragraphs are disconnected and separated from one another.”

We have thoroughly revised the introduction to align it more clearly and assertively with the manuscript’s objectives, methodology, findings, and implications. The updated version follows a more structured progression: it begins by contextualizing the severity of water pollution in Colombia using recent, peer-reviewed evidence; it then outlines the institutional framework for water quality assessment (IRCA); and finally introduces the gap between objective indicators and public perception as a key issue within the broader context of water injustice.

Furthermore, the introduction highlights the relevance of studying perception by linking it to socio-economic and structural inequalities, referencing comparative literature and Latin American studies (e.g., Afroz et al., 2015; Capelari et al., 2024). It also details the analytical strategy, including the use of LASSO regression and double machine learning techniques, and underscores the study’s interdisciplinary contribution to the field of Political Ecology.

• “The literature review section requires streamlining and further interlinkage.”

In response, we have significantly revised and reorganized this section to enhance clarity, thematic focus, and logical flow. The literature review is now divided into four interlinked subsections:

1. Environmental justice, which provides the conceptual foundation and introduces multidimensional frameworks (e.g., Schlosberg, Sen, Fraser) and their relevance to distributive, procedural, and restorative approaches.

2. Water justice, which contextualizes environmental justice in the domain of water, highlighting sociopolitical and infrastructural exclusion, and integrating emerging concepts such as hydraulic citizenship and intersectionality.

3. Manifestations of water injustice: international evidence, which examines regional case studies from Latin America, as well as global patterns (e.g., Portugal, Finland, Kenya).

4. Factors influencing perceptions of water quality, which integrates global and regional to explain how demographic, and environmental factors shape subjective evaluations of water quality.

We have also improved internal transitions and removed redundancies to ensure greater fluidity and integration across subsections.

• “The methods section also requires assertiveness and literature support no listing.”

We have revised the methodology section to adopt a more argumentative and explanatory tone, replacing descriptive listings with a theoretical justification for each methodological decision. In particular, bibliographic support was incorporated for both the selection of machine learning algorithms (LASSO and Double Machine Learning) and the predictors included in the analysis. Relevant references, such as Friedler et al. (2021), Yektansani et al. (2024), and Koroleva et al. (2020), were included to support the use of these techniques in social and environmental analysis contexts. Additionally, a conceptual justification was added for each variable category, explaining how sociodemographic, environmental, subjective health, pro-environmental practices, and infrastructure-related factors influence perceptions of water quality, based on empirical literature.

• “The absence of references before equations would mean complete originality, clearly not the case.”

Before each equation, a direct reference to the original author of the method used was included: Tibshirani (1996) for the penalized logistic regression (LASSO) and Chernozhukov et al. (2018) for the Double Machine Learning (DML) approach. This ensures the proper attribution of the analytical frameworks and prevents any misunderstanding regarding their authorship.

• “The results section requires significant improvements, upper letter no to be used, why add % next to each value_ Etc.”

The results section has been revised to ensure clarity and consistency. Uppercase letters were removed from non-standard titles or descriptors, and the use of the percentage symbol (%) has been limited to the column header and explanatory note, avoiding its repetition next to each value.

• “It is not clear what is being presented.”

To address this concern, a concluding paragraph was added at the end of the section, clearly stating what is being presented and why. This paragraph clarifies that the analysis aims to reveal water injustice in Colombia by highlighting the gap between public perception and IRCA objective indicators. The figures allow us to contrast reported perceptions with technical data, revealing a consistent pattern of underestimated risk in vulnerable settings

• “Why add columns where the value never changes?”

We reviewed the table and removed any columns where values remained constant across strata, as they did not add analytical value.

• “The discussion section would have to be merged with the results section supporting or opposing the results presented.”

We revised the manuscript to integrate the discussion directly with the results section. The revised structure links each empirical finding to relevant scholarly literature, either supporting or contrasting the results with existing research.

• “The conclusions section would have to start by clarifying the context, the methods used, the main findings, the implications, the limitations and future prospects.”

In response, we have thoroughly revised the Conclusions section to ensure a clear and structured presentation. The revised section now:

• Clarifies the context of the study by introducing the water quality perception gap in Colombia;

• Describes the methodology, highlighting the use of national survey data and advanced techniques such as Double Machine Learning (DML) and LASSO regression;

• Summarizes the main findings, including the disconnect between IRCA and public perception, and the role of socioeconomic and environmental factors;

• Outlines key implications for water governance, communication strategies, and community engagement;

• Identifies limitations such as the lack of data from riverine and Indigenous communities;

• Proposes future research directions, especially the integration of qualitative and participatory methods in marginalized regions.

• “This is clearly not the way to indicate data used as authors would have to further specify.

Availability of data and materials. All data used in the paper are publicly available at https://microdatos.dane.gov.co/index.php/catalog/793/get-microdata”

We have revised the “Availability of Data and Materials” section to provide a more specific and appropriate citation.

---

## [Editor Report · Decision Letter 1]

29 Aug 2025

Water Injustice in Colombia: Perceptions and Realities of Water Quality Examined Through Advanced Machine Learning

PONE-D-25-04781R1

Dear Dr. Lis-Gutiérrez,

We’re pleased to inform you that your manuscript has been judged scientifically suitable for publication and will be formally accepted for publication once it meets all outstanding technical requirements.

Kind regards,

Diogo Guedes Vidal, PhD

Academic Editor

PLOS ONE
---

## [Editor Report · Acceptance letter]

PONE-D-25-04781R1

PLOS ONE

Dear Dr. Lis-Gutiérrez,

I'm pleased to inform you that your manuscript has been deemed suitable for publication in PLOS ONE. Congratulations! Your manuscript is now being handed over to our production team.

Kind regards,

on behalf of

Dr. Diogo Guedes Vidal

Academic Editor

PLOS ONE